# Use of Computed Tomography and Magnetic Resonance Angiograms Combined with a 3D Surgical Guide in an Elderly Cat with an Occipital Lobe Meningioma

**DOI:** 10.3390/vetsci10040264

**Published:** 2023-03-29

**Authors:** Pillmoo Byun, Yoonho Roh, Haebeom Lee, Jaemin Jeong

**Affiliations:** 1College of Veterinary Medicine, Chungnam National University, Daejeon 34134, Republic of Korea; 2Institute of Animal Medicine, College of Veterinary Medicine, Gyeongsang National University, Jinju 52828, Republic of Korea

**Keywords:** angiography, meningioma, craniotomy, feline, 3D printing

## Abstract

**Simple Summary:**

Meningioma treatments, including complete surgical resection, debulking, radiation therapy, and palliative care in dogs and cats have recently been improved. Although complete resection of meningiomas is known to be the most effective treatment, there are still challenges in performing safe surgeries due to the presence of numerous vessels surrounding the tumor and brain parenchyma. Many efforts have been made in the field of human medicine to achieve safety, such as preoperative planning using 3D simulation and contrast angiography. Surgical navigation systems have been studied for application, but recently, three-dimensional techniques have helped surgeons perioperatively. This study enhanced the vessels around the meningiomas in the skull and attempted to identify them using 3D computed tomography and magnetic resonance imaging angiography rather than conventional angiography. An 11-year-old castrated male cat was referred to our hospital with a meningioma located in the occipital lobe. The cat showed progressive tetraparesis. Throughout the long-term follow-up, following the complete removal of the meningioma, the cat showed favorable clinical outcomes and no neurological abnormalities.

**Abstract:**

We present a case of occipital lobe meningioma resection in an elderly cat. The surgery was performed with the goal of avoiding major bleeding. An 11-year-old castrated indoor-only male Persian Chinchilla (5.5 kg) was presented with a month-long history of progressive tetraparesis for a left occipital lobe meningioma. Magnetic resonance imaging revealed a T2-weighted heterogeneously hyperintensity and a T1-weighted well-contrast enhancing extradural mass in the left occipital lobe of the brain. Cerebral angiographic data were obtained using magnetic resonance (MRA) and computed tomography angiography (CTA). Advanced angiograms and virtual reconstruction of images revealed that the tumor was surrounded by the caudal parasagittal meningeal vein. A left caudal rostrotentorial craniotomy and en bloc resection of the tumor were performed, and histopathology revealed a meningioma. Complete neurological recovery was achieved within 10 days after surgery. To the best of our knowledge, this is the first case report describing CTA and MRA findings and favorable clinical outcomes after surgical management of a brain meningioma without severe perioperative complications.

## 1. Introduction

Feline intracranial meningiomas are mainly benign extra-axial brain tumors that grow from the meninges and can be attached to the dura mater; they have been reported to occur frequently in primary intracranial neoplasia [1]. Intracranial meningiomas are more frequently diagnosed in elderly felines because the clinical signs are often mild and nonspecific. Among the brain regions, the third ventricle (17.2%) was the most frequently affected site, followed by the parietal lobe (15%), frontal lobe, and occipital lobe [1,2]. Understanding the prevalence of affected regions is essential because patients with meningiomas often show clinical signs related to the affected region [1,2].

There are various treatment options, such as chemotherapy and radiotherapy, although surgical resection is reported to be the most effective [2,3,4]. Since several regions can be affected, diverse surgical approaches are designed to expose meningiomas in different lobes [5]. Among various locations, occipital lobe surgery carries a high risk of complications, such as visual impairment or injury to vital structures, such as the transverse and occipital sinuses [2,5,6,7,8,9]. Consequently, it could be an additional risk factor for patients of advanced age, as they are more vulnerable to intraoperative blood loss [10]. Since blood loss affects the surgical outcome of resection, it is essential to accurately identify the vasculature of the occipital lobe and the meningioma [10,11,12]. A small amount of blood loss can cause infarction perioperatively, requiring prolonged surgical and anesthesia time to control bleeding during surgery [3,10,11,12,13,14,15,16,17]. In addition, the restricted working space in brain surgery makes it difficult to control bleeding effectively [18]. Thus, patient selection for surgical resection should be executed carefully because it is uncertain whether the elderly can recover from neurological deficits after surgery, let alone given the high risk of surgery in this population [7,8].

Diverse advances, such as minimally invasive techniques, approaches, and surgical devices, have been developed to overcome the challenges of brain surgery [19]. Furthermore, better diagnostic tools are required to preoperatively assess intracranial lesions and their association with surrounding tissues [15,20]. Tremendous efforts have been made to develop advanced diagnostic imaging techniques in human medicine [13,20,21,22,23,24]. Several methods have been developed to assess the blood vessels of meningiomas (feeding artery), including intra-arterial digital subtraction angiography (DSA), computed tomographic angiography (CTA), and magnetic resonance angiogram (MRA) [13,20,21,22,23,24]. While DSA, a catheter-based angiography, remained the standard by having the highest sensitivity and specificity in determining sinus patency and identifying feeding arteries of intracranial meningiomas, the cost and the risk for developing neurological complications decreased utilities compared to the less invasive modalities, such as CTA and MRA [24,25]. Regardless of the pros and cons of each method, angiography prior to the resection of intracranial neoplasia helps visualize the relationship between blood vessels and neoplasia [13,20].

However, to the best of our knowledge, there are no published reports on the use of advanced imaging angiograms for surgical planning of occipital lobe meningiomas in cats. This case report describes the application of CTA, MRA, and 3D printing techniques in the surgical planning of intracranial meningiomas in elderly cats, primarily to avoid injury to major brain vessels for better surgical outcomes.

## 2. Case Presentation

### 2.1. Case

An 11-year-old castrated indoor-only male Persian Chinchilla (5.5 kg) was referred to our hospital for left occipital lobe meningioma. The patient had a history of progressive tetraparesis that lasted for six months prior to the presentation. Two weeks prior to the presentation, the patient showed altered mental status due to anorexia and was admitted to a local hospital. The patient had a tonic-clonic seizure for 15 s during hospitalization, and no seizures were observed after discharge. The primary veterinarian had prescribed anticonvulsants because of the seizure, including gabapentin (10 mg/kg PO twice daily), phenobarbital (3 mg/kg PO twice daily), and levetiracetam (15 mg/kg PO twice daily). The day after discharge, the patient was referred to a local veterinary imaging center and admitted for magnetic resonance imaging (MRI). The patient was referred to our hospital with a suspected left occipital lobe meningioma. On presentation, the abnormal physical findings included ataxia and lethargy. Abnormalities observed duirng the neurological examination included obtundation and symmetrical cerebellar pelvic limb ataxia. However, the other neurological findings were normal. The patient was assessed with a score of 16/18 according to the Modified Glasgow Coma Scale, which falls into the Score III category with a good prognosis [26]. There were no difficulties with urination or defecation noted by the owner. In addition, the menace response test, pupillary light reflex, and visual follow-up showed that the patient’s vision was intact. A complete blood count was performed, and mild thrombocytopenia (128 K/uL, reference range: 156.4–626.4 K/uL) was observed. The serum panel, including FSAA, was unremarkable, and the ProBNP level was normal.

### 2.2. Images

General anesthesia was induced with propofol (4 mg/kg IV) and maintained with inhaled isoflurane and oxygen. The MRI sequences included 3-plane scout localizers, T1-weighted (TR/TE = 576/14 on transverse and 556/14 on sagittal planes), T2-weighted (TR/TE = 4600/90 on transverse, 2380/81 on sagittal, and 3000/81 on dorsal planes), FLAIR (TR/TE = 8000/77 on transverse plane), and postcontrast T1-weighted (TR/TE = 576/14 on transverse and 556/14 on sagittal and dorsal planes) images with a slice thickness of 2 mm. The MRI revealed a round, solitary, well-defined mass with a homogenous hypointensity when compared to gray matter in T1-weighted images, dura tail signs in T1-weighted contrast images, and edematous changes around the occipital lobe tumor in T2-weighted images. There was no sign of hyperostosis in the bone surrounding the mass (Figure 1).

The patient’s cerebral angiographic data were obtained using the computed tomography (CTA) and magnetic resonance angiography (MRA) after the MRI scan (Figure 2). After the scanning procedure, the patient recovered without complications. CTA was performed using a 64-channel multidetector scanner (Toshiba Aquilion; Toshiba, Tochigi, Japan) with parameters of 0.5 mm section thickness, 1.0 mm interval, 0.4 s rotation time, 0.641 pitch, 120 kilovoltage peak, and 120 milliampere-seconds. The patient was placed in a sternal recumbency position. A contrast medium injector (EmpowerCTA^®^+ Injector System, Bracco, Italy) was attached to the patient via a 24-gauge intravenous catheter in the cephalic vein. First, a precontrast scan was performed to define the location of the carotid artery for bolus tracking. After defining the location of the common carotid artery, a 2.5 mL/kg dose of iodinated contrast medium (600 mg iohexol/kg) was injected at a 1.5 mL/s injection rate. Hounsfield unit of the common carotid artery was monitored, and scans were manually initiated immediately after opacification of the artery to obtain an arteriogram. A venogram scan was started 30 s after the arteriography. On CTA, the transverse sinus was intact (Figure 2A), yet the mass compressed a vein suspected to be a caudal meningeal vein (Figure 2B). In addition, there was no evidence of invasion into the cerebellum in the supratentorial region, although lysis of the ipsilateral tentorium cerebelli was observed. MRA scanning was performed using a 3.0 Tesla magnetic resonance scanner (INGENIA, Philips Healthcare, Amsterdam, The Netherlands). Various sequences were used for visualization of the cerebral vasculature due to the small-sized nature of the object: 3D time of flight (TR/TE/NSA = 23/3/1, matrix size 500 × 333, field of view (FOV) 200 × 200), phase contrast (TR/TE/NSA = 12/7.6/1, matrix size 188 × 167, FOV 160 × 160), MRA (TR/TE/NSA = 4/2/1, flip angle 20°, matrix 384 × 381, FOV 210 × 210, acquisition time of 1 min 40 s, 0.5 slabs, voxel size 0.65 × 0.65, parallel imaging factor 0.65), and contrast-enhanced MRA (TR/TE/NSA = 4/2/1, matrix 384 × 381, FOV 210 × 210) sequence (Clariscan^TM^, GE Healthcare, Oslo, Norway) (0.1 mmol Meglumine gadoterate/kg). Signals from the cerebral arteries were not identified on the MRA. Additionally, no evidence of neoplastic invasion of surrounding vessels was observed. The cerebral arteries and veins were manually segmented using the original and post-processed datasets. The virtual reconstruction of the brain was performed using an image-processing program (MeVisLab; Fraunhofer Mevis, Bremen, Germany) (Figure 3) [27].

Bone data were obtained from the CT images and sliced into an STL file using 3D slicer software (3D Slicer, [www.slicer.org] accessed on 24 November 2021). With the MR images and virtual images of MRA, rehearsal surgery was performed using 3D modeling software (3DS MAX; Autodesk, Inc., San Rafael, CA, USA) to determine the operation window in order to minimize surgical damage to the skull and surrounding tissues. In addition, a virtual 3D surgical guide was designed using 3D modeling software (3DS MAX; Autodesk, Inc., San Rafael, CA, USA) to cover the bony surface of the window for the exact location during the operation (Figure 4). The virtual guide was transferred to an STL file and printed using an FDM-type 3D printer and a PLA filament [27].

### 2.3. Surgical Techniques

The information obtained from CTA and MRA confirmed that the tumor had not invaded the dorsal sagittal sinus or the caudal meningeal vein. Despite the slight displacement of the cerebral vein, as shown in the angiograms, the intact vascularity ensured well-defined capsulation of the meningioma and an en bloc resection was decided.

The surgical procedures were performed with informed consent from the owner and followed the Animal Use and Ethics Guidelines from the CNU University. The patient was premedicated with midazolam (0.2 mg/kg IV), anesthetized with propofol (4 mg/kg IV), and maintained with isoflurane. Intraoperative analgesia was provided by a constant rate infusion (CRI) of remifentanil (0.1–0.3 μg/kg/min). The patient was positioned sternally and recumbently, and the head was slightly tilted so that the meningioma was positioned at the highest point [6]. All the surgical procedures were performed under a surgical microscope (Leica M530 OHX; Leica Korea, Seoul, South Korea). The approach technique was extrapolated from a human occipital craniotomy [28]. To create the operative window, a horseshoe incision was made according to the preoperative plan. Prior to craniotomy, a patient-specific 3D surgical guide was placed over the skull to confirm the location of the lesion [27]. Once confirmed, a craniotomy line was marked with a monopolar electrocautery along the window created by the 3D surgical guide. Craniotomy was performed using a Sonopet bone saw (Stryker, Kalamazoo, MI, USA) (Figure 5A) [27]. As the bone flap was elevated, the dura was opened, and a meningioma was identified (Figure 5B). As observed in the preoperative MRA and CTA, the venous sinus was intact and isolated from the meningioma. The meningioma was removed using a Sonopet soft tissue aspirator (Stryker, Kalamazoo, MI, USA) with an en bloc resection (Figure 5C,D).

A small portion of the tumor was submitted for biopsy to a pathologist (IDEXX Laboratories, Inc., Westbrook, ME, USA). Hemostatic products, such as regenerated cellulose (Surgicel^®^, Ethicon, Somerville, NJ, USA), gelatin sponge (Spongostan^®^, Ethicon, Somerville, NJ, USA), and Floseal (Floseal^®^, Baxter Healthcare Corporation, Fremout, CA, USA) were used to manage bleeding (Figure 6A). An artificial dura (Redura, Medprin Biotech, La Mirada, CA, USA) and a dural sealant (Duraseal, Confluent Surgical Inc., Waltham, MA, USA) were used for dural closure (Figure 6B,C). To prevent iatrogenic metastasis, the inner table of the bone flap was drilled out before replacement [29,30]. Two craniofacial 1.6 plates (a rectangular four-hole plate, an X-shaped six-hole plate, and 4 mm self-tapping screws; JEIL, Seoul, South Korea) were used for bone flap fixation (Figure 6D). Finally, the skin and scalp were closed in a routine pattern.

### 2.4. Postoperative Care, Outcomes, and Pathological Examination

Immediately after surgery, the patient was sedated for 24 h with medetomidine (1 μg/kg/h CRI) to prevent excitement. All vital signs were normal and the appetite came back on postoperative day two. On postoperative day nine, gait improved, yet mild ataxia in the right hindlimb was still present. No other neurological deficits were clinically observed. Ten days after the surgery, the patient was discharged.

Histopathological examination revealed a grade 1 intracranial mass, a psammomatous subtype, and a meningioma (Figure 7). The results confirmed that the neoplastic cells predominantly displayed a whirling configuration with overwhelming numbers of psammoma bodies, consistent with the psammomatous subtypes. Pleomorphism was mild to moderate, and mitotic activity was low (1 mitosis/10 HPF) and compatible with grade 1 (low-grade) meningiomas. It was not possible to assess the marginal or angiolymphatic invasion due to the incisional nature of the sample [31]. However, we grossly identified the complete resection of the tumor. No complaints related to the surgery were received until postoperative day 445, when the owner sent a video of the patient showing full recovery from ataxia without neurological problems.

## 3. Discussion

We present a case of occipital lobe meningioma resection with the goal of avoiding major bleeding in an elderly cat. To minimize hemorrhage during surgery, preoperative CTA, MRA, and virtual 3D rehearsals were performed, and a 3D printing surgical guide was used. Therefore, the surgery was successfully performed without significant blood loss, and the patient completely recovered from the surgery despite its concerning older age.

Surgery involving the occipital lobes may have significant postoperative risks, ranging from neurological deficits to surgery-related deaths [11,12,32,33,34,35]. To minimize these risks, accurate presentation of blood vessels near surgical sites has been advocated in human medicine [35,36]. However, there are no reports on cranial blood vessel visualization in surgical planning in veterinary medicine. The vasculature of the occipital lobe in cats includes several major vessels, such as the dorsal sagittal sinus, transverse sinus, caudal dorsal cerebral vein, and more [37]. These vasculatures must be considered in surgical planning because surgical approaches involving these critical blood vessels are associated with high mortality and morbidity [3,38]. Furthermore, highly vascularized tumors, such as meningiomas, often result in significant hemorrhage during resection [13]. Thus, the accurate vascular status of tumors, such as occipital lobe meningiomas, must be identified to avoid the catastrophic outcome of the surgical operation [24,32]. In this case, using the preoperative CTA and MRA image data, the surgical guide was designed to avoid adjacent vessels while securing an adequate surgical window to excise the tumor completely. In addition, the guide guaranteed shorter operations and anesthesia times, which led to improved surgical outcomes [27,30,38,39,40,41]. Although the surgical approach to the occipital lobe was technically challenging, the angiograms and 3D-printed surgical guides aided the overall surgical procedure, resulting in a better outcome.

In human medicine, various efforts to mitigate blood loss during resection, including accurate assessment of the vasculature and preoperative embolization, are utilized for patients with intracranial tumors [13,22,42]. These efforts are based on current studies regarding the complications of intraoperative blood loss during intracranial tumor surgery [10,11,12]. However, in veterinary medicine, the effects of these advancements on patients with meningiomas have not been thoroughly studied, hence further research is required. Therefore, as the first attempt to report the use of CTA and MRA in a feline patient with an occipital meningioma, we described the imaging techniques in detail. There were several considerations when applying CTA and MRA to this patient. First, the concern that iodinated contrast medium may affect the image quality of MRI could be ignored because the purpose of the scan is to investigate the anatomical structure of intracranial vessels, and not the pathologic status of the brain lesion [43]. Additionally, because of the small size of the patient, the cerebral vasculature in the initially obtained image dataset could not provide sufficient information for surgical planning. Thus, CTA and MRA images were registered, and various digital subtraction methods were used (e.g., subtraction of pre-contrast CT images from arteriogram CT images to obtain bone-free arteriograms) to offset this limitation [24]. The combination of CTA and MRA offers a high spatial resolution of the tumor’s vascularity.

Preoperative images could provide information regarding the meningioma and its neighboring vasculature in detail. Thus, the surgical techniques applied in this case were mainly focused on minimizing iatrogenic damage to the surrounding blood vessels, such as the parasagittal sinus. Several advanced techniques, that were extrapolated from human medical procedures, have been applied in this case [15,16,33,39,44,45]. First, a horseshoe incision and occipital craniotomy were used in this case to expose the meningioma [16,46]. An occipital craniotomy is a versatile approach to the occipital lobe, that not only offers adequate exposure to brain lesions, but also provides a safer surgical method (i.e., dynamic retraction) to prevent the damage of the visual field cortex [8,33]. Furthermore, hemostatic products, including Surgicels and Floseal, were applied in this case to control intraoperative bleeding [45]. Although Floseal has some controversial aspects in neurosurgery, particularly in terms of thrombotic complications, using it in combination with Surgicel, as reported in the literature, has resulted in improved gelatin granule coherence in this case [44]. Additionally, a previous study reported that the application of Floseal showed decreased scar tissue formation and dural adhesion in the canine laminectomy model [47].

This study had several limitations. First, this single case does not represent all elderly feline patients with occipital meningiomas. Further studies should be performed to characterize feline occipital meningiomas, particularly in geriatric populations. Moreover, due to the small size of the patient, arterial angiography could not be performed. Thus, the feeding arteries of the meningioma may have been overlooked in this case. Additionally, we did not perform a follow-up CT or MRI postoperatively to assess nerve tissue adaptation and local adaptive or pathological changes. Simple clinical evaluation does not fully represent the extent of tumor removal, as well as changes that occur in surrounding tissue following surgery. Lastly, the deleterious effects of applied angiograms have been reported at low rates in human medicine [23]. Although the prevalence of side effects was insignificant, further studies should be conducted to evaluate the harmful effects on dogs and cats to advocate the use of angiograms.

## 4. Conclusions

In this study, by using advanced angiography techniques, such as CTA and MRA, we defined the displacement of peritumoral vessels in the preoperative plan. Although these angiographies are not routinely used in veterinary medicine, understanding the intracranial vasculature is essential for better surgical outcomes, especially in elderly patients prone to complications related to blood loss during surgery. To make surgical resection a feasible choice with a guaranteed acceptable outcome in elderly patients, an appropriate assessment of intracranial vascularity should be considered.

## Figures and Tables

**Figure 1 vetsci-10-00264-f001:**
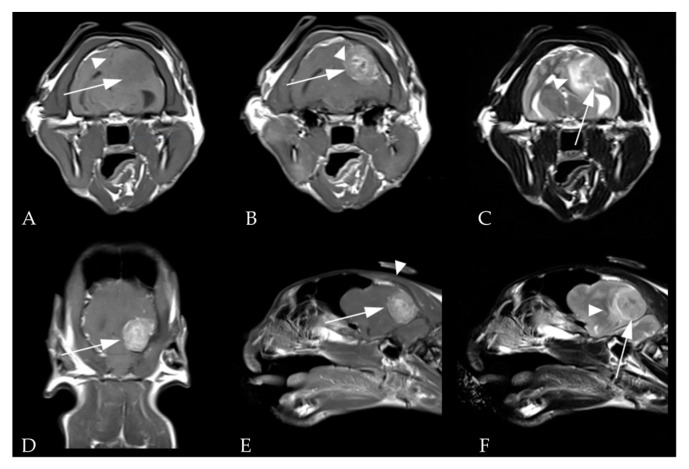
Preoperative magnetic resonance images (MRI) of the brain. (**A**) Transverse view of the brain in a T1-weighted window showing the neoplasm (arrow) with a homogenous hypointensity compared to gray matter. The cerebral midline is shifted (arrowhead). (**B**) Transverse view in a T1-weighted window after contrast enhancement revealing a dural tail sign (arrowhead) dorsal to the mass (arrow). Transverse (**C**) and sagittal planes (**F**) in a T2-weighted window showing edematous changes (arrowhead) around the tumor (arrow). Mass (arrow) is homogenously hyperintense in the dorsal (**D**) and sagittal views (**E**) in a contrast enhanced T1-window.

**Figure 2 vetsci-10-00264-f002:**
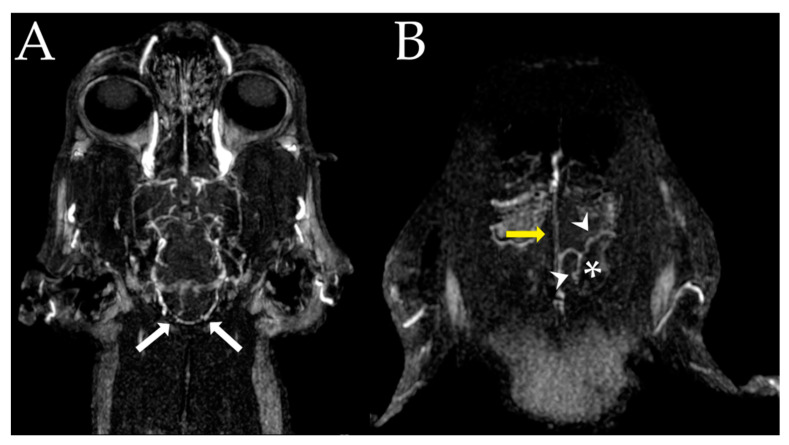
Advanced angiogram images. The patient was positioned in sternal recumbency. (**A**) On the axial view, the transverse sinuses (white arrows) are not affected by the mass. (**B**) The dorsal sagittal sinus is intact (yellow arrow) although the caudal meningeal vein (arrowheads) is observed as displaced by the mass effect exerted by the meningioma (asterisk) on the axial view.

**Figure 3 vetsci-10-00264-f003:**
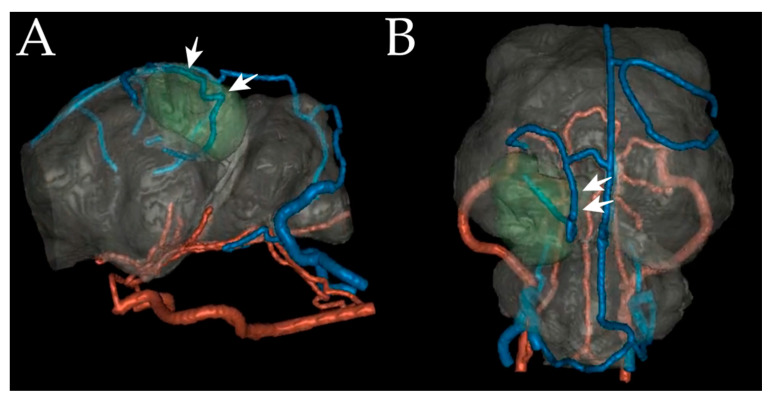
Virtual reconstruction images of the left lateral (**A**) and top view (**B**). The meningioma, depicted as green, is surrounded by the caudal meningeal vein (arrows).

**Figure 4 vetsci-10-00264-f004:**
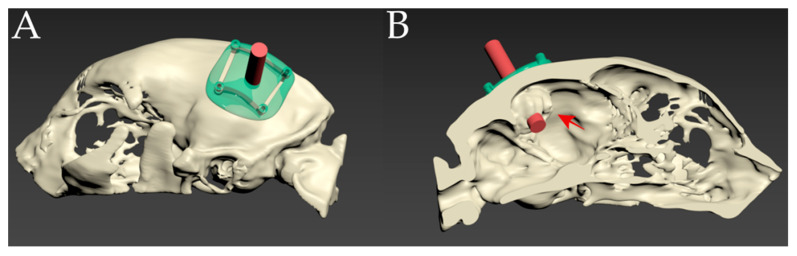
(**A**) Schematic images of the pre-operative skull-contoured three-dimensional (3D) patient-specific surgical guide. (**B**) The tumor (red arrow) is confirmed by a 3D bone model with a sagittal cut of the skull. A 3D tumor-targeting red cylinder passed the center of the tumor.

**Figure 5 vetsci-10-00264-f005:**
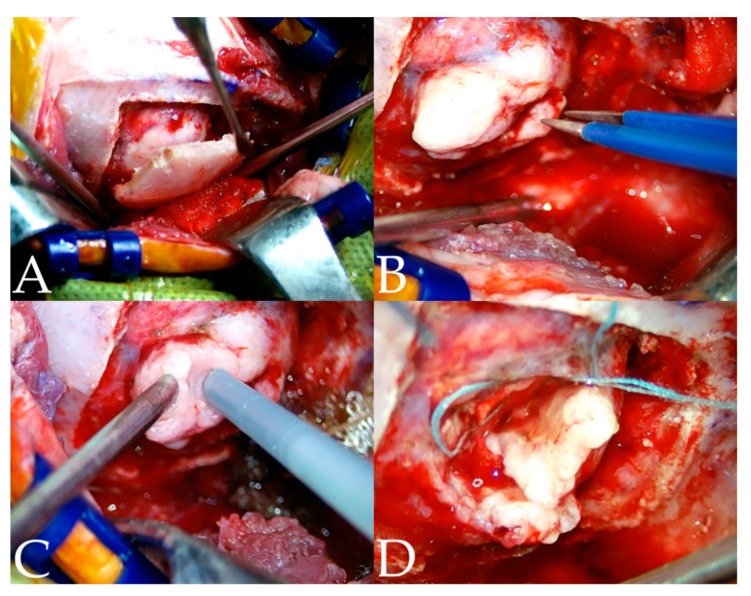
(**A**) Opening of the cranium. (**B**) The meningioma was identified, and bipolar was used to mitigate adjacent vessels during the resection. (**C**) The mass was removed using the ultrasonic aspirator in order to reduce the size of the mass. (**D**) The tumor was isolated from the brain parenchyma using neurosurgical sponges.

**Figure 6 vetsci-10-00264-f006:**
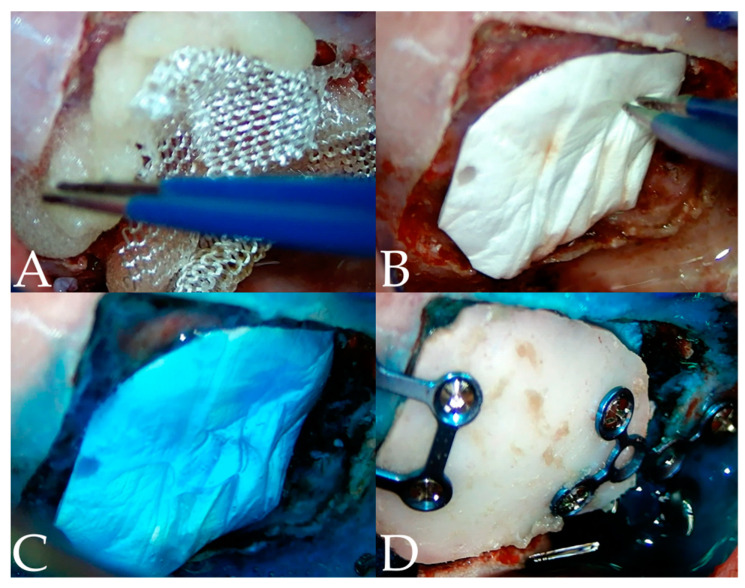
When minor hemorrhage occurred after the resection, Floseal and Surgicel (**A**) were used at the surgical site. (**B**) Once the bleeding stopped, artificial dura was used to cover the defect of the meninges. (**C**) The dural sealant was used over the artificial dura to prevent the leakage of CSF. (**D**) The osteotomy block was placed on the defect and fixed with a neurosurgical plate.

**Figure 7 vetsci-10-00264-f007:**
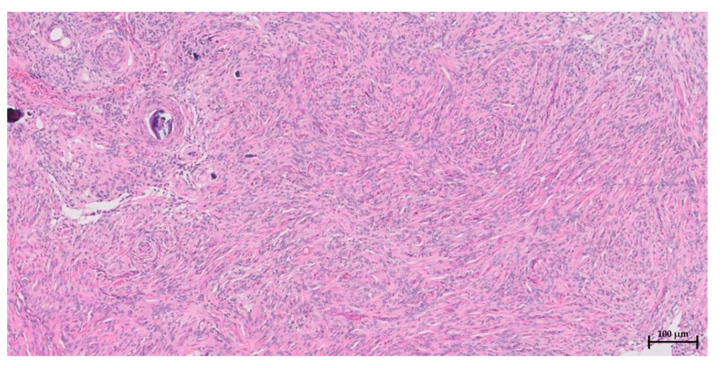
Microscopic picture of the incisional sample of the meningioma. It is densely packed with neoplastic cells supported by a fibrovascular stroma. Cellular nuclei are oval with finely stippled chromatin, with mild to moderate anisocytosis and anisokaryosis. The sample was diagnosed as the psammomatous subtype of meningioma with grade 1 (low grade) (H&E staining, scale bar = 100 µm).

## Data Availability

Not applicable.

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
