# Peer review of "Use of Computed Tomography and Magnetic Resonance Angiograms Combined with a 3D Surgical Guide in an Elderly Cat with an Occipital Lobe Meningioma"

_vetsci, 2023, doi:10.3390/vetsci10040264_

Round 1

Reviewer 1 Report

Use of computed tomography and magnetic resonance angiograms combined with 3D surgical guidance in an elderly cat with an occipital lobe meningioma

the case report is very interesting, well expounded and original both in terms of the diagnostics employed and the planning and execution of surgical treatment. The surgical location of the meningioma makes treatment particularly complicated and risky, this adds positive value to the importance of description and disclosure.

The keywords employed are appropriate.

The discussion is explanatory and comprehensive. The conclusions are consistent and agreeable

However, there are some critical issues that should be noted.

First, the most important limitation of the description, moreover identified and correctly reported by the authors in the discussion, is the absence of a diagnostic examination (MRI) post-surgery. 

For the type of surgery, MRI in the postoperative period is indispensable and very useful at a distance to assess nerve tissue adaptation and local adaptive or pathological changes. Simple clinical evaluation, at a distance, does not appear proportionate to what has been done to address and resolve the problem. 

I would suggest that the authors describe more fully in the discussion the consequences of this limitation.

A few more observations in the text

Line 55-56: the sentence seems unclear, I suggest rephrasing it

Line 148: what does "guardian" mean? perhaps owner?

Line 172: Figure 3. Images are too small and low resolution. I would suggest to the authors, even by splitting the figure into two or three, to include photos of larger size and at appropriate resolution to make it better for the reader to understand. The same concept, to the extent possible, also applies to Figure 2

Reviewer 2 Report

Review Byun et al 2023 Veterinary Sciences

Thank you for sharing your experience with feline intracranial meningioma. I think your article is interesting, but please get an English editing service to correct your article. Here are some of my suggestions.

Line

15 In human medicine. A lot of.. sentence

17 delete during surteries

17 surgical navigation systems have been

Line 19-20 delete We started…

22 After removing whole meningioma .. English

25 was presented

31 it is not a temporal but a caudal rostrotentorial approach in my opinion

42 delete than canines

Line49-52 ? English

Line 55/56 english and what do you meen al little loss of blood can make the infarction? I do not understand

57 … brain surgery makes surgery difficult

Line 58 should be made carefully

Line 64 remove to reduce the surgical complication in human medicine

Line 114 sternal recumbency

Figure 2 larger and better labelling, also the different views are not correct, the vessels should be labelled accordingly, better resolution possible?

Figure 3 Discuss spreading of tumor cells with replacement of bone flap

194 delete the owner was

Figure 4 bar is missing (any other special stains)211

Reference 35 is this the correct one? Does not fit with the message

Line 220 guide on the images?

226 printed surgical guide English

Line 248 expose convexity meningiomas … ? Line 253-255

Picture of the surgical guide is missing, what material has been used?

Please have an english editing service correcting this article,

 could be shorter

Reviewer 3 Report

The sent-for-review article “Use of computed tomography and magnetic resonance angiograms combined with 3D surgical guide 3 in an elderly cat with an occipital lobe meningioma” describes a single case of a brain tumor and the authors try to answer a question if the use of MRA and CTA helps the surgeon limit the perioperative bleeding.

The purpose of the study is not stated correctly,  in my opinion. Maybe try something like “The purpose of this study was to investigate if the use of CTA and MRA improves the avoidance of major bleeding”? Also, I would expect to read about some pros and cons of each imaging technique or their comparison.

The simple summary, abstract, and introduction are written in poor English and need a rewrite. Some sentences are too short and lack a verb, and some are so long that the reader feels lost and confused. Later parts are generally well-written, but you can still find a few mistakes. I suggest the help of professional translators.

The figures used for this article are too small, and there are too few of them. Could you make them bigger and present better the pictures from MRA and CTA?

The article lacks technical specifics of the diagnostic imaging studies. Please supplement them.

Line 26-27:

There is a mistake: the found mass is, according to the pictures in fig. 1, heterogeneously hyperintense in T2-weighed images, but is definitely not hyperintense in T1-weighted images. The mass is also not well-circumscribed, its borders are well-seen only after the contrast enhancement.

Line 40-41:

Meningiomas don’t grow within the dura matter, they grow from the meninges and can be attached to the dura matter.

The paragraph that contains lines 47-60:

It is not clear why the authors discuss the problem of blood loss in meningioma surgery in human medicine. More so to do it in the introduction part. The authors describe one case of meningioma in a cat, so there is no explanation for using their outcome in human medicine. Maybe leave that part in the discussion section?

Lines 160-161:

I think there is a mistake, shouldn't it be “As observed in the preoperative MRA and CTA…”, instead of “MRI and MRA”?

In the Surgical technique part, the authors wrote about the various methods to stop the perioperative bleeding, such as Surgicel, Spongostan, and Floseal. This seems excessive for a method that was supposed to limit the bleeding to the minimum.

There is also no description of how the use of MRA and/or CTA changed the surgical approach in the “surgical technique” section, but the authors put it in the “discussion” section. I would suggest moving it to the “surgical technique” section instead.

Line 183-184:

The authors wrote that there were no neurological deficits, yet they mention mild right hindlimb ataxia. Maybe change it to “no other neurological deficits were observed”?

In conclusion, in my opinion, the topic described in the article is interesting and the work should be considered for printing after corrections, especially in terms of language.

Round 2

Reviewer 1 Report

Dear authors,

thank you for sharing my suggestions and updating the manuscript.

Reviewer 3 Report

The article has been corrected as suggested by the reviewer and should be accepted for publication.